# Determinants of Alcohol Consumption among Medical Students: Results from POLLEK Cohort Study

**DOI:** 10.3390/ijerph18115872

**Published:** 2021-05-30

**Authors:** Maksymilian Gajda, Katarzyna Sedlaczek, Szymon Szemik, Małgorzata Kowalska

**Affiliations:** 1Department of Epidemiology, School of Medicine in Katowice, Medical University of Silesia, 40-055 Katowice, Poland; mkowalska@sum.edu.pl; 2School of Medicine in Zabrze, Medical University of Silesia, 40-055 Katowice, Poland; katarzyna.sedlaczek@wp.pl; 3Department of Nursing Propaedeutics, School of Health Sciences, Medical University of Silesia, 40-027 Katowice, Poland; sszemik@sum.edu.pl

**Keywords:** hazardous alcohol drinking, harmful alcohol drinking, alcohol consumption, medical students, AUDIT, AUDIT-C

## Abstract

Background: The use of alcohol is a serious public health concern all over the world, especially among young people, including students. Medical students are often exposed to higher levels of distress, which may lead to a higher prevalence of psychoactive substance use and psychiatric co-morbidities. Alcohol abuse can be one of the detrimental methods of coping with distress. The aim of this study was to assess the prevalence of alcohol use among medical students in Poland. Methods: We analyzed data from the POLLEK cohort study on alcohol consumption and possible influencing factors. Results: Among the 540 students included, 167 (30.9%) were hazardous drinkers (HAZ) according to the AUDIT test. The main identified risk factors of hazardous/harmful drinking were male gender and smoking cigarettes. Conclusions: Given the fairly widespread alcohol abuse among medical students, it is necessary to implement screening (and intervention in the next stage) programs in these groups.

## 1. Introduction

The use of alcohol is a growing public health concern all over the world; it is associated with risk factors for disability, preventable mortality, and crime [1,2,3,4,5,6,7,8]. It is a rising problem especially among young people, including students [2,3,4,5,7,8,9,10,11]. Their alcohol intake is higher than that of their peers not attending universities, as well as the general adult population [2,5]. It has been estimated that one in three people who abuse alcohol in their youth will become addicted [10]. As many as 9% of teenagers worldwide may be alcohol-dependent [3]. In the Netherlands, 24% of students are hazardous drinkers compared to 10% in the general adult population [2]. Similar negative behaviors have been observed among medical students who are often exposed to higher levels of distress compared to the age-matched population. In this group, alcohol misuse can be one of the detrimental methods of coping with academic, social, and economic stressors [3,5,8,10,12,13,14]. However, other researchers have suggested that medical students may not be more stressed compared to business students. They have also referred to studies with opposite results, presenting both lower and higher levels of stress among medical students [15]. Researches from the UK have reported students of the first and second years of medicine to be at lower risk of substance misuse than law students [16].

It has been already reported that alcohol misuse may have a negative short-term impact on academic performance, including dropout as well as long-term risk of alcohol use disorder (AUD) [2,5,12,17]. In spite of this, it is often underestimated, and even socially acceptable, when taken to be typical student behavior of limited duration [2,12]. On the contrary, other researchers argue that alcohol abuse by college students should not be considered a temporary problem [18]. Formed at such a young age, drinking habits may persist after graduation. These circumstances can delay the identification of those at risk and lead to long-term consequences [2,10]. Unexpectedly, medical students may receive proper counseling less often than the general population [9], and be hesitant to seek help and formal consultation [15]. Moreover, the knowledge about the negative health impact of alcohol may be on a low level even among medical students [6,7]. That is why the early identification of those at risk is so important.

Furthermore, the problem of alcohol also affects physicians [3,5,19,20]. Starting such a demanding job, which medical students prepare for, may even increase their exposure to stressors [10]. A French study revealed that alcohol misuse may be more frequent among “front-line” doctors practicing more stressful specializations (like emergency medicine or surgery) [21]. It may negatively influence their health status as well as their career, including counseling patients with substance use disorders [5,12,21,22]. Some physicians openly express their concerns about the impact of alcohol abuse on the quality of their work [21].

The highest level of alcohol consumption per capita is observed in Europe [2]. According to the Organization for Economic Cooperation and Development (OECD), alcohol was responsible for 11% of all deaths from preventable causes in the European Union in 2017. It was estimated that from 250,000 to 290,000 deaths across the EU are due to the harmful consumption of alcohol [1]. According to the World Health Organization (WHO), in 2016, the 12-month prevalence of alcohol dependence and harmful use among Poles aged 15 or over was higher, while alcohol dependence was less frequent, than in the whole WHO European Region (12.8% vs. 8.8% and 2.2% vs. 3.7%, respectively). Alcohol attributable deaths were mainly due to liver cirrhosis, road traffic injuries, and cancers, with male predominance. It resulted in high a Alcohol-attributable Years of Life Lost (YLL) score of 4 (out of 5) for Poland [23]. Experts from the OECD and the European Observatory on Health Systems and Policies prepared Country Health Profiles, covering all of the European Union (EU), as well as Norway and Iceland. The latest report is “Country Health Profiles 2019”. According to this report, alcohol could be attributed to 7% of all deaths in Poland. Alcohol consumption is higher than the EU average. Moreover, it has increased over the last fifteen years in contrary to the falling tendency in the EU.

Given the above, the aim of the presented study was to assess the prevalence of alcohol use among medical students attending two medical faculties (in Katowice and Zabrze) of the Medical University of Silesia in Katowice (MUoS).

## 2. Materials and Methods

### 2.1. The Study Design and Sample

A total of 638 first-year medical students from two faculties (Katowice and Zabrze) of MUoS were invited to participate in a longitudinal, observational study named POLLEK. Finally, 540 participants (84.6% of all invited) for AUDIT and 544 participants (85.3%) for AUDIT-C with complete data were included in the analysis. Data collection was performed during classes, both on paper as well as (for a large number of students) using an online form (via a dedicated webpage created for the POLLEK study). It was fully anonymous, except for gathering students’ e-mail addresses in the electronic part. Students participated entirely voluntarily and they did not receive any financial compensation. Each participant gave informed written consent. We used the authors’ questionnaire, which among other assessments of socio-demographic and health behavior data also included gender, height, weight, blood pressure (BP), body mass index (BMI), marital status, smoking status, basic eating habits, and ever diagnosed chronic diseases. Alcohol intake was assessed with the AUDIT (Alcohol Use Disorders Identification Test) tool, which is covered in more detail later in this section [24]. Very good accordance and reproducibility of paper and electronic versions of the POLLEK questionnaire were demonstrated in our previous paper [25]. The presented results of the current cross-sectional study are based on data obtained in the academic year 2019/2020.

### 2.2. Instruments

As stated earlier, we used the AUDIT questionnaire to estimate the prevalence of hazardous drinking among medical students. It is a gold standard screening 10-item tool for hazardous drinking in the adult population developed in 1989 by WHO [2,24]. It encloses questions covering patterns of alcohol use, its negative effects, and signs of addiction. The first part of AUDIT (questions 1–3) measures the frequency and amount of drinking, while the second part (questions 4–6) assesses alcohol-related problems, and the last part (questions 7–10) evaluates alcohol-related mental and physical problems. Responses to items range from 0 to 4 points (except for the last two questions, with possible scores of 0, 2, or 4). Therefore, the range of possible scores is 0–40. AUDIT-Consumption (AUDIT-C), a shorter version of AUDIT, was evaluated almost as accurately as the original test in the adult population [26]. AUDIT-Consumption (AUDIT-C) is an abbreviated version of this reliable and simple test. AUDIT-C is composed of the first three questions of AUDIT, measuring the frequency of alcohol use. Moreover, to assess participants’ quality of life, we used the Polish version of the WHOQOL-BREF questionnaire, which was developed by WHO [27]. It consisted of 26 items grouped into four domains: somatic, psychological, social, and environmental. In our previous publication, we proved the legitimacy of using this tool in our group [25].

### 2.3. Measures

Hazardous (HAZ) drinking of alcohol was identified in the subjects with a score of ≥8 in AUDIT, as this value was most frequently used in previous studies [4,5,8,10,16,17,21,28,29,30]. The problematic consumption or dependence was determined as a score of ≥16 [21,28]. The design of the POLLEK study includes a prospective assessment of the study group in the following years (current results will be considered baseline). However, we are aware that due to the COVID-19 pandemic, the implementation of these plans may be difficult. Additionally, to compare the results with other researchers frequently using the AUDIT-C scale, its values were also assessed. AUDIT-C is composed of the first three questions of the full AUDIT. Scores to each item may range from 0 to 4 points, resulting in a final score from 0 to 12. We adopted >4 as the cut-off value for AUDIT-C, as is the frequently used value in former studies [3,6,7,14,21].

### 2.4. Statistical Analysis

After collecting data, we performed simple descriptive analyses. Given the non-normal distribution of quantitative variables assessed by the Shapiro-Wilk test, nonparametric statistics (χ^2^, Fisher exact, and U Mann-Whitney tests) were used. Categorical variables were presented as numbers and percentages, while median and interquartile ranges described continuous variables. We described sociodemographic characteristics and the prevalence of hazardous drinking as median values (Me) and interquartile ranges (IQR) of AUDIT/AUDIT-C for the whole study group, and selected groups defined by gender, faculty, BMI, smoking status, and other selected factors. Spearman’s test was used to compute correlations between AUDIT scores and particular categorical or quantitative variables. Moreover, to identify the characteristics and profile of hazardous drinkers, univariate analysis and multivariate linear regression analysis were used. Their results were reported classically as regression coefficients with 95% confidence intervals (CIs). We conducted preliminary analyses to ensure that assumptions of linear regression were not violated (including normality, linearity, multicollinearity and homoscedasticity). Given the high correlation between variables related to smoking, we decided to exclude e-cigarettes from the final model. After applying the Akaike Information Criterion (AIC) in a stepwise algorithm (backward direction), another two variables were removed: physical activity and BMI. All analyses were performed in R 4.0.3 software [31], while the level of significance was defined at a *p* < 0.05 criterion.

### 2.5. Ethical Approval

The ethics approval for the POLLEK study was received from the Bioethical Committee of the Medical University of Silesia in Katowice (approval number KNW/0022/KB/217/19, date: 8 November 2019). Written informed consent was obtained from all participants.

## 3. Results

### 3.1. Sociodemographic Characteristics

The median (Me) age of respondents was 19 years (IQR: 19–20). Among them, 332 (61.5%) were female, 385 (71.3%) single, and almost all (517, 95.7%) were financially dependent on their parents. More than half (58.3%) had never smoked cigarettes, and 24.8% of students had quit smoking, while only 14.3% were active smokers. Only 84 respondents (15.6%) had a BMI denoting overweight, while the majority (56.1%) were normal-weighted and about one in three (27.8%) were underweight (missing data for three students).

### 3.2. AUDIT Scores

The median score on the full AUDIT for all medical students was 5 (IQR: 2–9). Based on the AUDIT scores, the majority of students were in a low-risk group with a median of 3 points (IQR: 2–5), 153 students were identified as hazardous drinkers (Me = 10; IQR: 9–12), only 9 were identified as harmful drinkers (Me = 17; IQR: 17–18), and only 5 might be addicted to alcohol (Me = 21; IQR: 20–21). Similar results were obtained based on AUDIT-C (also presented in Table 1 and Table 2). The full version of the analysis from Table 1 and Table 2 was included in Appendix A.

### 3.3. Factors Related to Hazardous Drinking

Students who were identified as hazardous drinkers were more often declared to be economically dependent on their parents (98.8% vs. 94.4%; *p* = 0.02). Smoking (current or former) was more prevalent among risky drinkers compared to the low-risk group (62.9% vs. 32.2%; *p* < 0.001). We also observed a similar significant difference in the prevalence of current smoking compared to never/former smoking (*p* < 0.001).

Hazardous drinkers have significantly lower scores in the BREF psychological domain compared to low risk participants (20 vs. 21 points; *p* = 0.03), and lower scores (however statistically not significantly different; *p* = 0.06) on the whole BREF scale (79 vs. 81). Similar results were obtained for the physical domain of BREF (18 vs. 19 points; *p* = 0.08), and no differences were shown for the social domain. Similarly, there was no relationship between the frequency of alcohol consumption and ever diagnosed chronic disease, nor with self-esteem of health status (data not included in Table 1). However, the subjects who remained in constant control due to a diagnosed chronic disease used alcohol significantly (*p* = 0.03) less often. 

In addition, we found a weak positive correlation between AUDIT score and systolic blood pressure (rho = 0.14; *p* < 0.001), BMI (rho = 0.12; *p* = 0.004) and a very weak negative correlation between values of AUDIT and BREF psychological domain (rho = 0.085; *p* = 0.049). There were not any significant correlations between AUDIT scores and other variables, including BREF score.

Univariate analyses have identified eight variables to be associated with hazardous alcohol drinking (Table 3). Finally, using a multivariate regression model to predict determinants of alcohol abuse, we identified the following two risk factors: male gender and current or former tobacco smoking (Table 3).

## 4. Discussion

Generally, a high prevalence of psychoactive substance use among medical students, including alcohol, has already been reported. It can be also associated with burnout syndrome, depressive disorders, and other psychiatric comorbidities [8,9,13,14]. Healthcare professionals are often seen by patients as role models for a healthy lifestyle. It is worth emphasizing that their health behavior (including alcohol consumption) may affect the quality of counseling patients [5,12,22]. Some countries have proposed or even introduced special programs preventing students and healthcare professionals from consuming alcohol [3,5,8,29]. Such initiatives enable public health experts to identify students in need of appropriate interventions, which need to be organized by decision-makers. Binge drinking among Polish adolescents is another cause for concern [32]. They are many terms describing risky alcohol use, including “binge drinking”, which most often denotes consumption of five or more drinks on one occasion [2,33]. However, definitions of this term vary between studies. Some authors defined binge drinking as more than 6 drinks on any occasion during the last year, while others defined it as ≥4 and ≥5 standard drinks (for women and men, respectively) on a single occasion [5,34].

AUDIT is the current gold standard screening tool for hazardous drinking in the adult population [2,24]. Other simple screening tools for identification of potential problems with alcohol are CAGE (acronym from italicized words in the questionnaire: “cut-annoyed-guilty-eye”) and MAST (Michigan Alcohol Screening Test) [35,36]. AUDIT was developed by WHO as a 10-item test, also providing a framework to reduce or cease risky drinking, thus resulting in avoiding harmful consequences [37]. AUDIT may identify more problem drinkers than CAGE [17]. As defined in the WHO AUDIT guide, hazardous drinking is a pattern of alcohol consumption that increases the risk of harmful consequences for the user or others, leading to physical or psychological damage. Harmful use was referred to as drinking behavior resulting in consequences for physical and psychological health, as well as social consequences. Finally, alcohol dependence is “a cluster of behavioral, cognitive, and physiological phenomena that may develop after repeated alcohol use” [24].

### 4.1. Cut-Off Values for AUDIT

In the general adult population, 8 points is the cut-off point recommended by the “WHO AUDIT guide” to identify hazardous and harmful drinking with the AUDIT tool [24]. A higher prevalence of drinking among students than in the general adult population determined using the “classical” cut-offs may result in many false-positive results. Therefore, Verhoog et al. recommended the cut-off value for students to be ≥11 points [2]. Other countries have their specific cut-off values. For example, in a French version of AUDIT, a score to identify hazardous drinkers was established to be 6 or 7 for females or males, respectively, while a score of 13 points enables the identification of alcohol-dependency [29]. Similar studies were performed in the United States of America (USA), resulting in slightly different results [38,39]. The cut-off for AUDIT adapted for U.S. standard drinks (US-AUDIT) is lower than in the original AUDIT. The values to identify at-risk drinkers and likely alcohol use disorders were 5 and 13 for men and 6 and 8 for women, respectively [38]. Among Japanese students, optimal AUDIT cut-offs for moderate drinking were 5 points for men and 4 points for women [33].

### 4.2. Cut-Off Values for AUDIT-C

Scores of 3 points for females and 4 points for men on the AUDIT-C are considered optimal for identifying hazardous drinking or active alcohol use disorder (AUD) [3,37]. Apart from adults, it was also validated to use among students with the optimal cut-off of 7 and 8 points for female and male students, respectively [2]. In a study conducted in Finland among medical students at the University of Tampere, risky alcohol drinking was identified depending on AUDIT-C scores of 5 or 6 points for female and male students, respectively [12]. Among Japanese students, the score of 4 points in AUDIT-C was established as the optimal cut-off for moderate drinking (regardless of gender) [33]. However, AUDIT-C was reported to overestimate the prevalence of risky alcohol consumption compared to the original AUDIT recommended by WHO [21]. It is worth mentioning that Ketoja et al. demonstrated that the shorter test “AUDIT-3” at the cut-off of ≥2 points might be almost as valid as AUDIT-C in detecting risky alcohol use among men, but not women [12].

### 4.3. Comparison of Prevalence of Alcohol Consumption

The prevalence of hazardous drinking in our study group of medical students is in line with other studies, discussed further and summarized in Table 4 (the full version of this table is available in the Appendix A). To our knowledge, the presented study is the largest in Poland evaluating the hazardous alcohol consumption in medical students with both AUDIT and AUDIT-C tools. Our review of the literature indicates that one of the highest prevalence of drinking concerns Italy, with 85.5% of medical students, 77.4% of physicians, and “just” 63% of other healthcare-profession students [3]. For comparison, only 20% of students in the Netherlands were identified as hazardous and harmful drinkers; however, they were not medical students [2]. In the German study by Voigt et al., 62.8% of medical students and 82.5% of physicians were declared to drink alcohol regularly [22]. Given that this study was not based on the AUDIT tool, the direct comparison is not possible. In a Polish study validating the use of AUDIT, none of the medical students from the Medical University of Warsaw (*n* = 405) met the criteria for alcohol dependence, nor required advice or referral for treatment of AUD symptoms [4]. The prevalence of alcohol consumption outside Europe is also high. Out of 157 of the medical students from the National University of Asunción in Paraguay, nearly half (49%) met the criteria for alcohol misuse or dependence in AUDIT-C. Additionally, 43.9% of them may have burnout syndrome, while 38.9% had high probability criteria for major depressive disorder [14]. In Nepal, 281 students (47.75%) were alcohol users, including 90 (32%) classified as harmful users. Researchers have also established that alcohol use was significantly more frequent among the students of subsequent study years [13].

#### 4.3.1. Gender Differences

Researchers from Finland found that approximately every second male and every fourth female student of the University of Tampere were risky alcohol drinkers [12]. Hazardous or harmful drinking of alcohol is significantly (*p* < 0.05) more prevalent among male medical students in the USA, Germany, and the UK [8,17,22]. Drinking alcohol as a method of coping with stress was also more prevalent among male medical students from Korea [5]. Contrarily, female medical students from France drink alcohol more often than their male colleagues (35% vs. 22%), but were significantly (*p* = 0.002) less likely to be alcohol dependent (2% vs. 8%) [9].

#### 4.3.2. Smoking and Alcohol

Our results, like those of other authors, show that there is a link between alcohol consumption and smoking. In a group of Finnish students, the prevalence of smoking was 22.7%. Significantly more smokers were among male and female risky alcohol users compared to moderate alcohol users (24.1% vs. 6.2% and 21.1% vs. 4.1%, respectively) [12]. Similar results were reported for Italian medical students, non-medical students and resident physicians [3], as well as for medical students in Spain [40] and Poland [6]. Smoking medical students from Korea were over 2.7 times more likely to be binge drinkers compared to their non-smoking colleagues [5]. Contrarily, another Korean study showed no association between smoking status and alcohol consumption [18].

#### 4.3.3. Social Determinants

Results of the survey conducted in France (*n* = 171) indicated that preclinical medical students were statistically significantly (*p* < 0.05) more often risky or hazardous alcohol drinkers compared to clinical medical students (47% vs. 16%). Researchers have reported no significant differences in alcohol use in terms of students’ economic status or their parents’ occupation [29]. On the contrary, our results could suggest that students living with parents may be at a higher risk of abusing alcohol; however, the results of multivariate linear regression did not confirm such a relationship. Family history of alcohol may be another factor related to hazardous drinking [11,39]. However, other studies conducted in Poland (involving medical and dentistry students as well as young physicians) did not confirm such a relationship [10].

### 4.4. Multivariate Approach to Identify Factors Predicting AUDIT Scores

Using a multivariate logistic regression model we were able to link gender, smoking status, and faculty with hazardous alcohol use. Our results are partly in line with the Italian study, also using a multiple regression model, linked regular alcohol use with male gender, young age, lower BMI, active smoker status, coffee drinkers, and physicians or medical students (rather than other healthcare-profession students) [3]. Similarly, studies conducted in the USA and UK linked smoking status, among other factors, with a possible alcohol use disorder [8,16].

### 4.5. Limitations of the Study

Despite the relatively high participation rate of 85%, the influence of selection and volunteer biases cannot be excluded. Moreover, recall bias should be taken into account as some of the questions referred to past experiences (including the drinking of alcohol). It should be emphasized that head-to-head comparisons of results between our study and former studies are limited by different measures and cut-off values. This limitation has already been noted by Ketoja et al. [12]. When interpreting our results, it should be taken into account that some of the respondents may not have answered truthfully due to the sensitive nature of some of the questions. Moreover, AUDIT as any self-report screening test may not completely objective. It can yield both false-negative and false-positive results. Though we surveyed a reasonable sample of medical students from two cities, we cannot exclude that they are a different population compared to other medical universities in Poland. Therefore, there is a need for larger studies involving participants from other Polish medical academic centers.

## 5. Conclusions

The obtained results indicate fairly widespread alcohol abuse in the study group. They are generally consistent with other studies and legitimize the need for the implementation of permanent supervision (screening programs) in the field of alcohol use by students at medical universities. As the next stage, brief intervention programs should be introduced to help reduce this negative phenomenon.

## Figures and Tables

**Table 1 ijerph-18-05872-t001:** The analysis of alcohol use regarding selected personal characteristics—qualitative variables with numbers and frequencies (in brackets) and *p*-values (missing values are not shown).

Variable	Value	AUDIT (*n* = 540)			AUDIT C (*n* = 544)		
		Hazard Use *n* = 167 (30.9%)	Low Risk *n* = 373 (69.1%)	*p*	Hazard Use *n* = 188 (34.6%)	Low Risk *n* = 356 (65.4%)	*p*
Gender	Female	71 (21.4%)	261 (78.6%)	<0.001	78 (23.4%)	255 (76.6%)	<0.001
	Male	96 (46.2%)	112 (53.8%)		110 (52.1%)	101 (47.9%)	
Source of income dependent on parents	Yes	165 (31.9%)	352 (68.1%)	0.02	186 (35.8%)	334 (64.2%)	0.004
	No	2 (8.7%)	21 (91.3%)		2 (8.3%)	22 (91.7%)	
Cigarette smoker	Never	62 (19.7%)	253 (80.3%)	<0.001	76 (23.9%)	242 (76.1%)	<0.001
	Current/former	105 (46.7%)	120 (53.3%)		112 (49.6%)	114 (50.4%)	
E-cigarette smoker	Never	73 (20.4%)	285 (79.6%)	<0.001	88 (24.4%)	273 (75.6%)	<0.001
	Current/former	90 (51.1%)	86 (48.9%)		95 (53.7%)	82 (46.3%)	
Under surveillance due to chronic disease	Yes	19 (26.4%)	53 (73.6%)	0.03	23 (31.5%)	50 (68.5%)	0.99
	No	17 (50.0%)	17 (50.0%)		14 (41.2%)	20 (58.8%)	
Uptake of sport activities to improve physical fitness	Yes	143 (32.3%)	300 (67.7%)	0.2	160 (35.8%)	287 (64.2%)	0.2
	Not at all	24 (24.7%)	73 (75.3%)		28 (28.9%)	69 (71.1%)	
Number of meals containing animal protein	100% of meals	28 (40.6%)	41 (59.4%)	0.2	31 (44.9%)	38 (55.1%)	0.05
	75% of meals	85 (29.5%)	203 (70.5%)		104 (35.7%)	187 (64.3%)	
	Less often	54 (29.5%)	129 (70.5%)		53 (28.8%)	131 (71.2%)	
Consumption of fruit and vegetables	Daily (≥3 meals)	19 (20.7%)	73 (79.3%)	0.07	22 (23.9%)	70 (76.1%)	0.05
	Daily (≥2 meals)	88 (33.5%)	175 (66.5%)		101 (38.0%)	165 (62.0%)	
	Less often	58 (31.7%)	125 (68.3%)		64 (34.8%)	120 (65.2%)	
Body mass index (BMI) (kg/m^2^)	<20	32 (21.3%)	118 (78.7%)	0.001	34 (22.7%)	116 (77.3%)	<0.001
	20–25	96 (31.7%)	207 (68.3%)		113 (36.9%)	193 (63.1%)	
	>25	37 (44.0%)	47 (56.0%)		40 (47.1%)	45 (52.9%)	

**Table 2 ijerph-18-05872-t002:** The analysis of alcohol use regarding selected personal characteristics—quantitative variables with medians (Me) and interquartile ranges (IQR) in brackets and *p*-values (missing values are not shown).

Variable	AUDIT (*n* = 540)			AUDIT-C (*n* = 544)		
	Hazard Use *n* = 167	Low Risk *n* = 373	*p*	Hazard Use *n* = 188	Low Risk *n* = 356	*p*
Age (years)	19 (19–20)	19 (19–20)	0.1	19 (19–20)	19 (19–20)	0.3
Weight (kg)	70 (60–80)	62 (55.3–71)	<0.001	70 (61.2–80)	61 (55–70)	<0.001
Height (cm)	176 (168–182)	170 (164–176)	<0.001	176 (169–182)	169.7 (164–176)	<0.001
Body mass index (BMI) (kg/m^2^)	22.9 (2.4–24.7)	21.5 (19.6–23.6)	<0.001	23 (2.5–24.7)	21.4 (19.6–23.4)	<0.001
Diastolic blood pressure (mmHg)	73 (66–80)	75 (67–80)	0.2	74 (67.8–80)	75 (67–80)	0.7
Systolic blood pressure (mmHg)	120 (111–127)	119 (110–124)	0.01	120 (111.8–128)	118 (110–124)	<0.001
AUDIT score	10 (9–13)	3 (2–5)	<0.001	10 (7–13)	3 (2–5)	<0.001
Hazardous Alcohol Use domain	6 (5–7)	3 (2–4)	<0.001	6 (5–7)	3 (1–4)	<0.001
Dependence Symptoms domain	1 (1–2)	0 (0–0)	<0.001	1 (0–2)	0 (0–1)	<0.001
Harmful Alcohol Use domain	3 (2–5)	0 (0–1)	<0.001	2 (1–4)	0 (0–1)	<0.001
BREF score	79 (71–85)	81 (73–87)	0.06	80 (73–85)	80 (72.8–87)	0.7
BREF physical domain	18 (16.5–21)	19 (17–22)	0.08	19 (17–21)	19 (17–21)	0.9
BREF psychological domain	20 (18–23)	21 (19–23)	0.03	21 (18–23)	21 (18–23)	0.7
BREF social domain	12 (10–13)	12 (10–13)	0.3	12 (10–13)	12 (10–13)	0.5
BREF environmental domain	29 (26–31)	29 (26–31)	0.4	29 (26–31)	29 (26–31)	0.9

*Note: n*, number of participants; *p*, statistical significance.

**Table 3 ijerph-18-05872-t003:** Results of univariate and multivariate linear regression analysis.

Variable	Value	Univariate	Multivariate Linear Model	
		*p*	Coefficient (95% CI)	*p*
Gender	Female	<0.001	reference	
	Male		2.04 (1.36, 2.72)	***
Age	Continuous variable	0.07	−0.16 (−0.37, 0.06)	NS
BMI	Continuous variable	0.014	---	---
Systolic blood pressure	Continuous variable	0.001	0.02 (−0.01, 0.04)	NS
Diastolic blood pressure	Continuous variable	NS	---	---
Source of income dependent on parents	No	0.002	reference	
	Yes		−1.68 (−3.56, −0.20)	NS
Cigarette smoker	Never	<0.001	reference	
	Current/former smoker		2.86 (2.24, 3.48)	***
E-cigarette smoker	Never	<0.001	---	---
	Current/former smoker			
Uptake of sport activities to improve physical fitness	Yes	0.07	---	---
	Not at all			
Consumption of fruit and vegetables	Daily	NS	---	---
	Less often			
Observations (*n*) = 513; R^2^ = 0.24; Adjusted R^2^ = 0.23				

Levels of significance: *** *p* < 0.001; NS, not significant; *p* ≥ 0.1 in univariate analysis; *p* ≥ 0.05 in multivariate.

**Table 4 ijerph-18-05872-t004:** Review of selected studies assessing alcohol use among students and doctors.

Country [Reference]	Studied Group; Size	Prevalence			HAZ Cut-Offs	
		HAZ%	HAR%	SEX	AUDIT	AUDIT-C
Brazil [28]	S; 398	22.4	-	-	8	-
Finland [12]	S; 465	33.0	-	M	-	M: 6; F: 5
France [21]	P; 515	12.6	1.2	F	8	M: 5; F: 4
France [29]	S; 171	11.0	21	-	8	-
France [9]	S; 198	31.9	-	F	M: 7; F: 6	-
Germany [30]	S; 80	24.0	-	M	8	-
Italy [3]	S; 641	6.1	-	F	-	M: 4; F: 3
Korea [5]	S; 323	45.5	T: 13.6; M: 18.4; F: 7.6	M	8	-
Korea [18]	N; 922	44.6 ^B^	T: 9.5 ^H^; M: 11.2; F: 8.1	M	M: 9; F: 6	-
Nepal [13]	S; 588	47.8 *	15.3	-	-	-
Netherlands [2]	N; 5401	2.0	-	M	11	M: 8; F: 7
Paraguay [14]	S; 157	49 **	-	F	-	M: 4; F: 3
Poland [6]	S; 194	54.1	-	M	-	M: 5; F: 4
Poland [10]	S, N, P; 268	16.8	1.9	-	8	-
Poland [4]	S; 405	-	-	-	8	-
Poland [7]	S; 635	47.1	-	M	-	M: 5; F: 4
Poland [11]	N; 500	65.0	-	-	-	-
Spain [40]	S; 192	49.5	-	M	M: 9; F: 6	-
Sweden [15]	S; 408	-	17.2	M	11 ^&^	-
United Kingdom [17]	S; 244	47.1	-	M	8	-
United Kingdom [16]	S; 820	4.5	5.2	-	8	-
United State of America [8]	S; 2710	18.1	-	M	8	-

HAZ, prevalence of hazardous alcohol users (percentages); HAR, prevalence of harmful alcohol users (percentages); SEX, gender predominance of alcohol risky use; S, population of medical students; N, population of other students; P, population of physicians; T, total group; M, male; F, female; -, not available; * alcohol users; ** alcohol misuse/dependence; ^B^ binge drinking; ^H^ heavy drinking; ^&^ cut-off for harmful alcohol use.

## Data Availability

The data presented in this study are available on reasonable request from the corresponding author. The data are not publicly available due to the data sensitivity and to protect the interests and privacy of the respondents.

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
