# Peer review of "Determinants of Alcohol Consumption among Medical Students: Results from POLLEK Cohort Study"

_ijerph, 2021, doi:10.3390/ijerph18115872_

Round 1

Reviewer 1 Report

The manuscript presents an interesting study on the prevalence of smoking and alcohol consumption in medical students from two Polish medical faculties.

It is a very relevant study given the present and future consequences of the degree of alcohol and tobacco consumption. Especially in future medical professionals, although it is limited by the specificity of the location of the sample, although this aspect is already referred to in the limitation section. It is deduced the importance of implementing specific intervention programs for the prevention and reduction of alcohol and tobacco consumption, especially in the first years, males, dependent on the parents' income.

The theoretical framework is up-to-date and is relevant to the object of study. The description of the participants is complete, as is that of the data analysis. The results are well described and correspond to the objective of the research. The discussion is extensive, complete, and well argued.

However, special care should be taken when drawing conclusions from the Source of income-dependent on parents, since the number of subjects who are economically independent is very small, especially those at high risk. Possible reasons for the differences between Katowice and Zabrze students should be specified with more detail (AUDIT, p = .03; AUDIT-C, p = .004)

Reviewer 2 Report

In this descriptive study the authors survey an impressive number of medical students and related demographic information to AUDIT scores. The manuscript is generally well written and I just have some minor comments.

  1. It is not clear to me why the authors need to report both the AUDIT and AUDIT-C results. Why not just use the full AUDIT seeing as the full AUDIT was used? In any case, it is confusing at section 2.2 when the AUDIT-C is introduced - it sets up the expectation to the reader that only AUDIT-C was used in this study.
  2. Section 3.1: 15% of respondents are overweight and 56% are healthy-weight - what about the other 30%?
  3. Weight and height are confounded with gender. So it doesn't make sense to correlate these separately with AUDIT, or include all these variables in the model. That is the function of BMI (to try and control for height in the weight distribution). So I would just use BMI.
  4. For any regression model I would expect to some basic assumption testing being reported. I would expect the authors to report which preliminary analyses were conducted to ensure that assumptions of normality, linearity, multicollinearity and homoscedasticity were not violated. I would expect the strongest intercorrelations, minimum tolerance values, maximum Mahalanobis distances and maximum Cook's distances to be reported. Did the authors examine the normal probability plots of the standardized residuals for major deviations from normality and a scatterplot of residuals for other violations of assumptions?
